# Evidence of Predation on Early Pleistocene Freshwater Ostracods (Umbria, Central Italy)

**Angela Baldanza [1,\*], Roberto Bizzarri [1], Francesco Posati [2] and Manuel Ravoni [3]**

[1] Department of Physics and Geology, University of Perugia, 06123 Perugia, Italy; roberto.bizzarri@libero.it
[2] Graduated in Natural Sciences, Via del Fagiano 3, 05100 Terni, Italy; francesco.posati@libero.it
[3] Geotechnical Laboratory, RPS Services UK & The Netherlands, Auriollaan 37, 3527ER Utrecht, The Netherlands; manuel.ravoni@rps.nl
\* Correspondence: angela.baldanza@unipg.it

**Abstract:** Although drillholes in modern and ancient ostracods are known, the record is relatively scarce when compared to other taxa, and mainly exist with reference to the marine environment. Moreover, less is known about perforated ostracods, and more generally, about bioerosion in freshwater environments. Traces of predation on freshwater ostracods are reported for the first time in deep-lake deposits belonging to the early Pleistocene Fosso Bianco Unit, and outcropping in the Cava Nuova section (Umbria, central Italy). Deposits are mainly clay to silty clay and sand; the fossil record is sparse, and is mainly comprised of very rare gastropods and bivalves, ostracods and plant remains (leaves, seeds and wood's fragments). The association of ostracods consists of *Candona (Neglecandona) neglecta*, *Caspiocypris basilicii*, *Caspiocypris tiberina*, *Caspiocypris perusia*, *Caspiocypris tuderis*, *Caspiocypris posteroacuta*, and *Cyprideis torosa*. The *Caspiocypris* group, considered to be endemic to the grey clays of the Fosso Bianco Unit, present the majority of specimens affected by predation, with a prevalence of predated female valves and a comparable number of right and left predated valves, while only a few of *Candona (N.) neglecta* (adult and juvenile) valves are perforated. Traces of predation for nourishment, represented by microborings of different types, were abscribed to the ichnospecies *Oichnus paraboloides* Bromley 1981, *Oichnus simplex* Bromley 1981, *Oichnus gradatus* Nielsen and Nielsen 2001, *Oichnus ovalis* Bromley 1993, and *Dipatulichnus rotundus* Nielsen and Nielsen 2001. Microboring affected both adult and juvenile specimens, evidencing prey–predator coexistence in the same environment over a long period of time. This report makes a fundamental contribution to the knowledge of predation in this peculiar confined environment, also suggesting prey–predator relations over a relatively short time interval (80–160 ka).

**Keywords:** predation; freshwater ostracods; microboring; early Pleistocene; deep-lake environment

## 1. Introduction

The term "bioerosion" was coined by Neumann [1] as an abbreviation of "biological erosion" and describes every form of biologic penetration into hard substrates, i.e., lithic (including skeletal) and woody. An extremely wide range of organisms cause bioerosion, and the work of these organisms produces trace fossils at all scales, from microscopic to gigantic [2]. Among others, Bromley [3] reconsidered this definition, adding "the process by which animals, plants and microbes sculpt or penetrate surfaces of hard substrate".

The term "microboring" refers to a trace left by another body or a mechanical abrasion trace, generally acting on mineralized shells. In any case, "perforation" indicates a phenomenon of interaction between organisms, or between organism and substrate. The most common drilling structures or anchors found in fossils or recent skeletal remains are "small depressions or holes of variable shape



from circular to sub-circular" [4]. These structures are generated by different organisms on a variety of skeletal substrates housing them [5,6], and they are generally made for predatory purposes, as described by several authors [7–13]. The perforations produced by predators on shells of prey organisms provide information on the predator–prey interaction, and are widely used by palaeontologists to quantify the organization of predation in the fossil record [14].

The ichnologist Richard Bromley [4] proposed the ichnogenus *Oichnus* for the circular perforations ("small round holes"), complete and incomplete, identified on calcareous shells; the author identified and described, in various geological records, the structures of small circular holes on valves and seashells, relatively uncommon in the Palaeozoic, but very common in the Cenozoic.

Recently, other authors [15] have revisited the taxonomy of the ichnospecies proposed for the "small round pits and holes" in fossil skeletal material found in a wide variety of invertebrates. Four ichnogenera have been proposed for these trace fossils: *Sedilichnus* Mueller, 1977, *Oichnus* Bromley, 1981, *Tremichnus* Brett, 1985, and *Fossichnus* Nielsen, Nielsen and Bromley, 2003. The identification of *Sedilichnus* as a new ichnogenus that groups together the characteristics of the "small round pits and holes" has not always been accepted, and ichnogenera *Oichnus* and *Tremichnus* have been revised. The ichnogenera *Balticapunctum* Rozhnov, 1989 and *Fossichnus* are considered to be synonymous with *Tremichnus* and *Oichnus*, respectively [16]. The refined ichnogeneric diagnoses return *Oichnus* to complete or incomplete bioerosive penetrations in calcareous skeletal substrates, commonly interpreted as praedichnia with or without signs of attachment. On the other hand, *Tremichnus* (now including *Oichnus excavatus*) exclusively refers to shallow pits passing into echinoderm skeletons that are interpreted as domichnia or fixichnia.

A complete list of all bioerosion ichnotaxa [17] recognized as valid, with the type ichnospecies name or combination and the type of bioerosion trace fossil, correlated with the nature of infested substrate and the known or inferred tracemaker, represents an essential overview for the research community and a tool for avoiding the uncontrolled proliferation of taxa that do not meet the definition of bioerosion trace fossils.

We refer to this list in our case study because the predation traces identified on the calcareous shells of invertebrates (fresh water ostracods) are in good agreement with the diagnosis and with the ethological categories of the new ichnofamily Oichnidae [17], and at least two of the ichnotaxa members have been recognized. These new data could help improve the basis for using bioerosion traces as palaeoenvironmental indicators.

*Microborings in Ostracods*

Studies on microborings mainly reports cases of predation on shells of both fossil and living marine brachiopods, bivalves and gastropods [18,19].

Nonetheless, research on perforations affecting the valves of ostracods, are very limited; contributions often refer to cases of predation in marine environments [20–23].

As for other taxa, two main kinds of holes have been commonly reported in marine ostracods [23]: (1) *Oichnus paraboloides* Bromley, 1981, and (2) *Oichnus simplex* Bromley, 1981. The predated valves usually show a single hole, although multiple perforations have been reported in a few cases. Type (1) presents paraboloid holes, elliptical or circular in outline, with an outside diameter (ranging from 50 to 500 μm) significantly larger than the inner. Outside the hole, a peripheral area with scratches and deep cuts is frequently present. Type (2) is characterized by cylindrical holes with internal and external openings of the same size (25–125 μm).

These two ichnospecies, particularly Type (1), are commonly attributed to predation by naticids [20,24] or other gastropods, such as Cassidae and Tonnidae [25], or the genus *Chicoreus*, and to parasitic strategies of Eulimids [26]. One example, among others, can be provided by the rare predation observed in ostracod populations collected in recent marine sediments of the southwestern Spanish shelf (0–70 m) [27]. In this case, the parabolic boreholes (*Oichnus paraboloides*) are dominant (>70%) over cylindrical boreholes (*Oichnus simplex*). Predation of the paraboloid type

(*Oichnus paraboloides*) predominates over cylindrical perforations (*Oichnus simplex*) in both marine and continental environments. The ornamentation of the valves does not affect the presence of perforations.

On the other hand, phenomena of bioerosion in freshwater environments have rarely been reported, and few cases of predation on ostracods have been documented, often in transitional environments.

An example of predation [28] was identified in existing populations of ostracods in two coastal lagoons, in semiarid North Africa (Morocco and Tunisia); specifically, in the innermost areas of both lagoons, where most of the perforated valves were found, belonging to euryhaline *Cyprideis torosa* and freshwater species *Heterocypris salina*, and more rarely to the marine species *Bairdia mediterranea* and *Bairdia longevaginata*. Each valve generally shows only a cylindrical hole, perfectly drilled, with inner and outer diameter of the same size and circular in shape. The hole diameter is 25 to 80 μm in *C. torosa* and *H. salina*, while it is greater in marine species (possibly reworked) *B. mediterranea* and *B. longevaginata*, with sizes between 100 and 150 μm. Most of the holes are located in the central part of the valve, near the inner muscle plaque. The holes were attributed to *Oichnus simplex*. In these lagoons, predation mainly affected adults and juveniles A-3 *C. torosa* and adults of *H. salina*, and the suspected possible predators were identified as the gastropods *Chicoreus* and/or parasitic Eulimids (some of which were identified in the bottom sediments of the lagoons), and/or small turbellarids [28].

The phenomenon of predation on populations of marine and continental Quaternary ostracods was also analyzed in the southwestern province of Buenos Aires (Argentina) [29]; out of 12,500 specimens, only 33 shells had been affected by predation, or in very low percentages. Predators mainly attached adults and the last juvenile stage. The diameters of the perforations ranged between 40 and 100 μm, with no direct relationship between the dimensions of the holes and the dimensions of the valves, and they were more frequently found in the anterior area of the valves. Based on morphology, the authors identified two types of perforations, *Oichnus paraboloides* and *O. simplex*, and concluded that most predation was associated with low-energy continental environments with a high influx of nutrients and with a greater density of ostracods [29]. Environments with higher energies show no valves with predation. Moreover, the researchers suggested that the factors influencing predation on ostracods are population growth, environmental energy, and nutrient availability [29].

Until now, data about predation on freshwater ostracods have been lacking; this work represents the first documentation of predated ostracods in the lacustrine environment, at least in the early Pleistocene. Thus, it sheds a new light on predation strategies in this peculiar confined environment.

In this case study, it was possible to identify a large number of predated valves and to recognize various ichnospecies of *Oichnus*, accompanied by rare *Dipatulichnus* and possible traces of anchorage, classified as fixichnia.

An open problem concerns the identification of possible predators: indeed, no fossil remains of carnivore gastropods were found, and the only inhabitants of the deep-lake able to fossilize were the ostracods.

## 2. Geological Setting

The study site is located in the present-day southern Tiber Valley (Umbria, central Italy, Figure 1). The area belongs to the Pliocene–Pleistocene South Tiberino Basin, one of the NW-SE oriented extensional basins characterizing the geological evolution of Northern Apennine from Miocene to recent [30,31]. Two main depositional cycles are recognized: (1) the lacustrine phase, Pliocene to early Pleistocene in age, associated with the informal stratigraphic units of Fosso Bianco (FBU) and Ponte Naja (PNU), and (2) the alluvial plain phase, early to middle Pleistocene, largely represented by the S. Maria di Ciciliano (SMCU) and, locally, by the Acquasparta (AU) units [32–35]. The study area lies inside the older cycle and belongs to the informal FBU Unit (Piacenzian–Gelasian) [32–35].

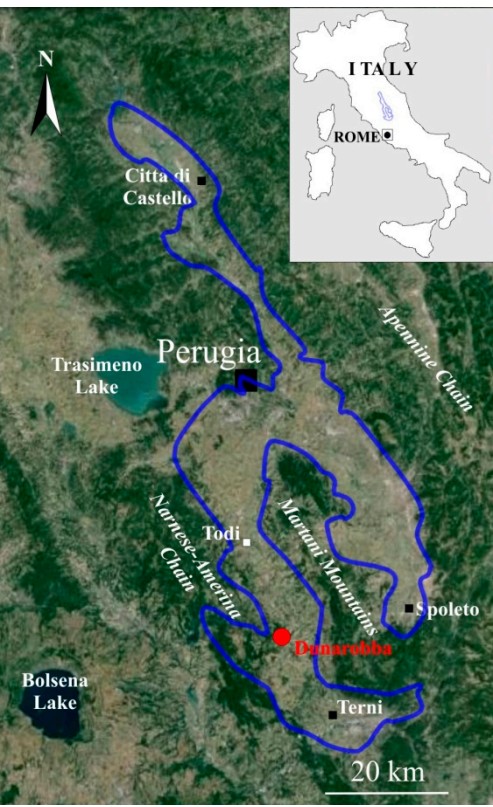

**Figure 1.** Geographic location of the Dunarobba site (Umbria, Central Italy) and the Cava Nuova section (red dot). The extension of the Tiberino basin during the Quaternary is outlined in blue. Modified from Google Earth image.

### 2.1. The Cava Nuova Section

The Cava Nuova section, inside the Fornaci Briziarelli Marsciano quarry, is the most accessible outcrop of FBU in the area. New lithological and sedimentological descriptions have recently been published [33,35], and we refer to these works for details and facies description.

The Cava Nuova (CND) section has a total thickness of 45 m (Figures 2 and 3); lithological and sedimentological features allow its division into three main intervals [32,35]:

- Interval 1 (from the base of the section up to meter 23): silty clay, massive to slightly parallel-laminated. Fossil record presents rare gastropods, ostracods, bivalves, plants (leaves and wood fragments). Thin horizons of lignite and lignite-bearing clay also occur, often marked by black or deep brown color. In the interval between meters 6 and 19, iron-enriched reddish horizons, $FeCO_3$ (siderite) crusts and nodules are common within the clay [36]. Deposits are related to a moderately deep lacustrine environment below the wave base. Inside this interval, the range between meters 4 and 22 is the only one containing ostracod valves affected by microboring.
- Interval 2 (from 23 m to 38 m): silty clay, alternating with fine sand and/or silty beds. Sands are usually ripple-laminated. Thin lignite horizons are commonly documented within the clay, as well as less frequent iron-enriched reddish horizons. Deposits are still referrable to a lacustrine environment, showing clear shallowing-upward trend, intermittent interaction with wave motion and resedimentation processes.
- Interval 3 (from 38 m to 45 m): alternation of sand and silt, from parallel-laminated to cross-laminated. Plant remains, frequently including leaves, are common, as well as lignite horizons. Deposits are associated with a lacustrine margin subject to wave motion.

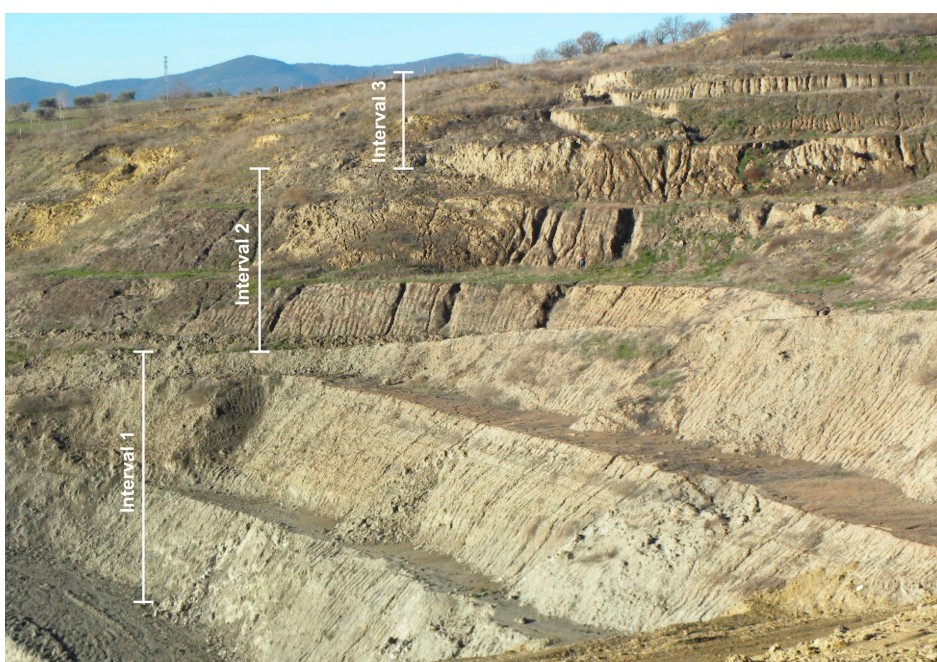

**Figure 2.** Overview of the Cava Nuova section and the limits of Intervals 1–3.

This coarsening- and shallowing-upwards trend has been recognized in the whole Dunarobba area [32,33,35], and has been interpreted in terms of progressive shift from deeper zones of the lake towards its coastal areas. Nonetheless, the arrival of sandy sediments and structures of wave ripples and dunes at 23 m (Figure 3) seems to mark an important change in the palaeoenvironment, mainly in terms of water energy.

Inside the long-lasting stratigraphic range of FBU, the study section represents a relatively short time interval. On the basis of the stratigraphic constraints [33,37], and of some speculation about the sedimentation rate, together with the comparison with other known successions with reference to the Fosso Bianco Unit, a time range between 2.5 and 2.2 Ma has been proposed for the whole Cava Nuova section [35]. In this scenario, the deep lacustrine deposits (i.e., the lowermost 23 m, belonging to Interval 1) could have been deposited in a time between about 80–100 ka and 155–160 ka, from about 2.5 Ma to about 2.34 Ma [35].

### 2.2. Fossils Content and Palaeoenvironmental Restoration

Throughout the section (Figures 2 and 3), the fossil record is quite sparse, presenting rare freshwater gastropods (*Emmericia umbra*), bivalves, and ostracods (Figure 4). The last-named belong to eight species [35]: *Candona (Neglecandona) neglecta, Caspiocypris basilicii* (Figure 4a,e,f), *Caspiocypris tiberina* (Figure 4c,h), *Caspiocypris tuderis* (Figure 4b,g), *Caspiocypris perusia, Caspiocypris posteroacuta, Cytherissa lacustris* (Figure 4d), and *Cyprideis torosa*. The five species of *Caspiocypris* dominate the assemblages recovered in samples of the Intervals 1 and 2. *Cyprideis torosa* prevailed within deposits of Interval 3, and *C. lacustris* occurred only in two samples from Interval 1 (Figure 3).

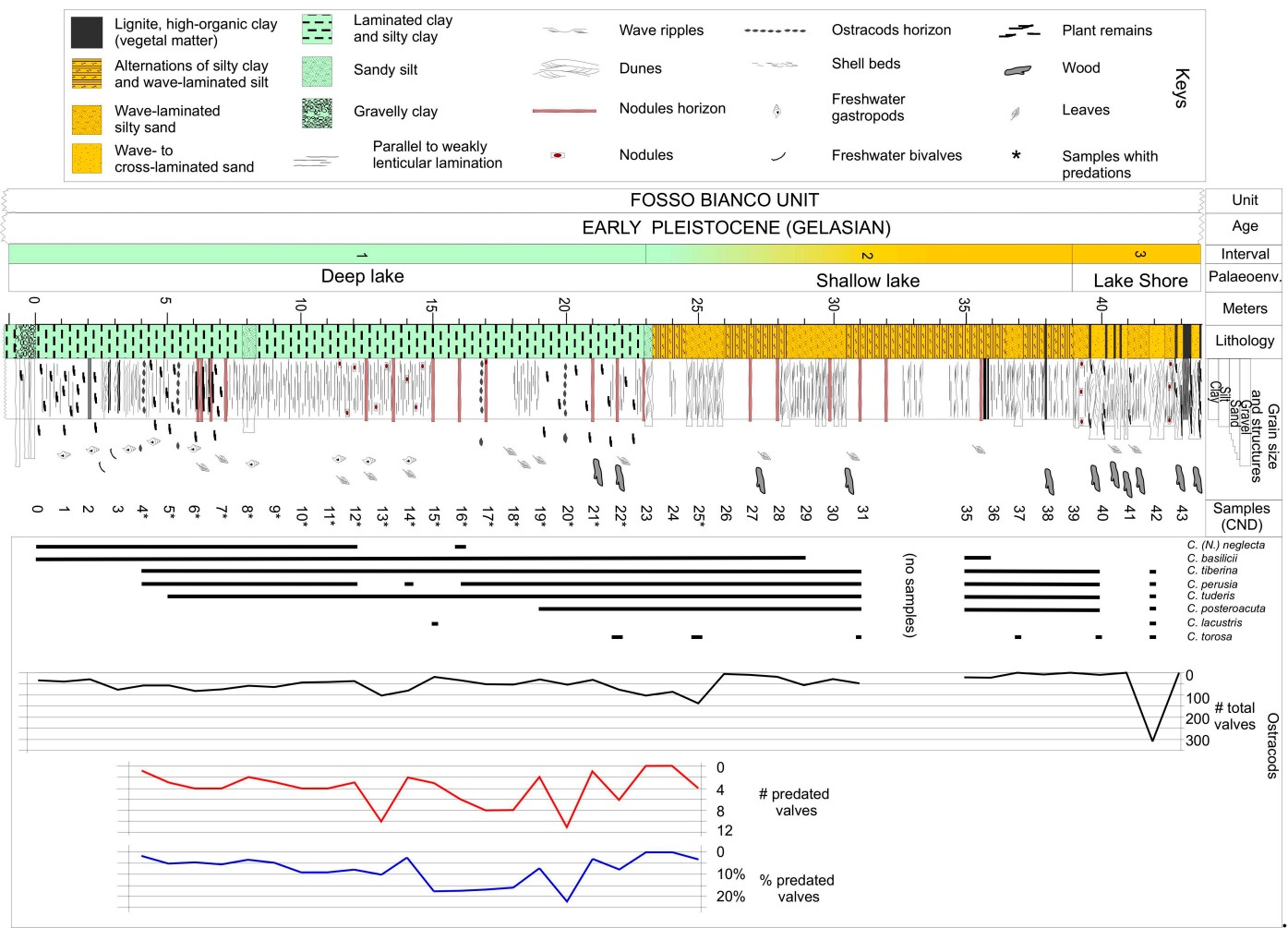

**Figure 3.** Sedimentological and stratigraphic log of the Cava Nuova section [34,35]. Main sedimentological structures, fossil content, distribution and frequency of ostracods (only disarticulated valves) and percentages of predated valves are reported.

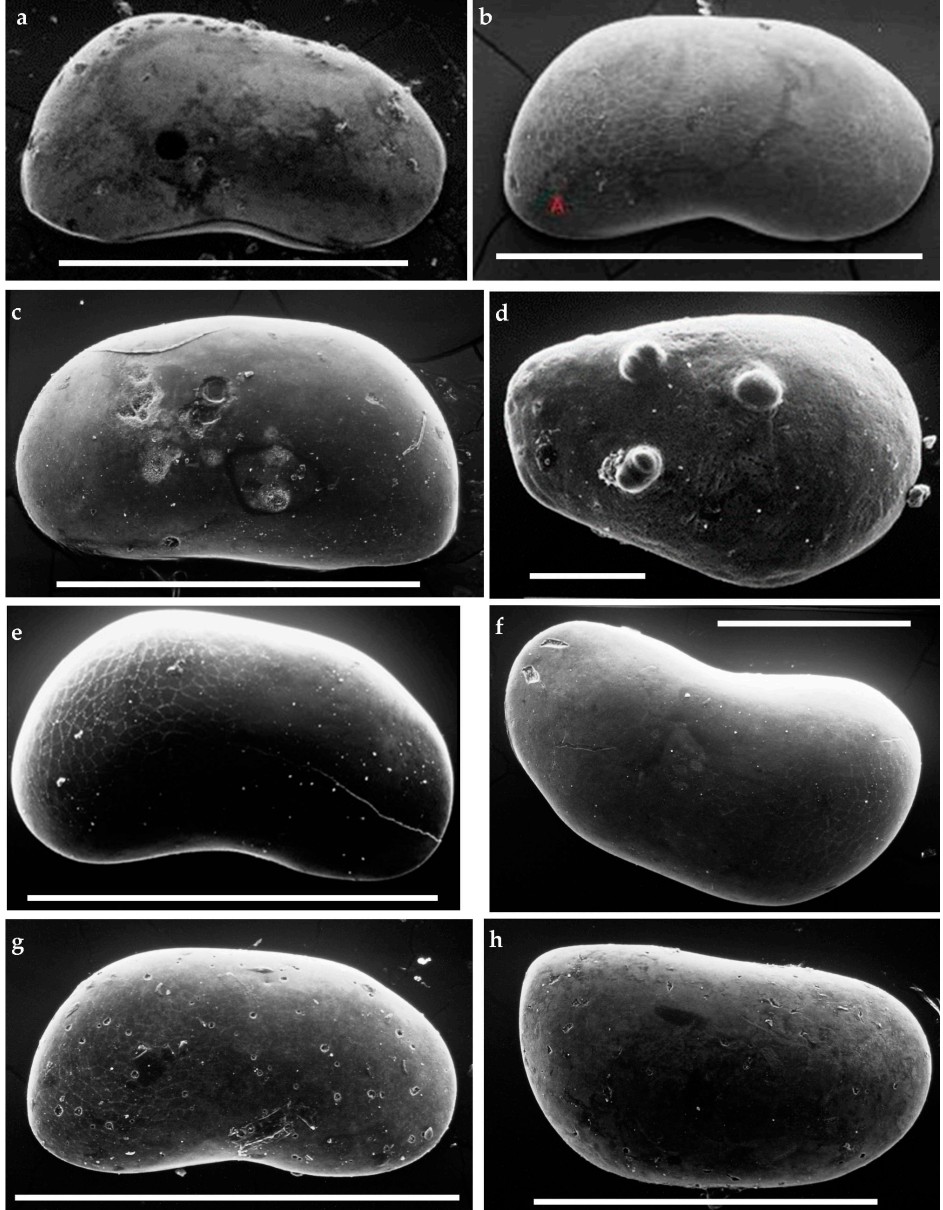

**Figure 4.** Scanning Electron Microscope microphotos of Cava Nuova ostracods, also showing microboring. (**a**) *Caspiocypris basilicii* Spadi and Gliozzi 2018, lateral view of right female valve, scale bar 1 mm. (**b**) *Caspiocypris tuderis* Spadi and Gliozzi 2018, lateral view of right female valve, scale bar 1mm. (**c**) *Caspiocypris tiberina* Spadi and Gliozzi 2018, lateral view of left female valve, scale bar 1 mm. (**d**) *Cytherissa lacustris* Sars 1863, lateral view of left valve, scale bar 200 μm. (**e**) *Caspiocypris basilicii* Spadi and Gliozzi 2018, lateral view of right female valve, scale bar 1 mm. (**f**) *Caspiocypris basilicii* juvenile right valve, scale bar 500 μm. (**g**) *Caspiocypris tuderis* Spadi and Gliozzi 2018, lateral view of right female valve, scale bar 1 mm. (**h**) *Caspiocypris tiberina* Spadi and Gliozzi 2018, lateral view of left female valve, scale bar 1 mm.

The five species of *Caspiocypris* recovered in this section are representative of a species flock [37]; *Caspiocypris* includes a group of closely related species characterized by monophyly, endemism and speciosity, confirming the "ancient-lake" nature of palaeolake Tiberino during the Piacenzian-Gelasian interval [38]. In this way, the long-lasting lacustrine sedimentation coupled with the occurrence of ostracods referrable only to Candoninae and Limnocytheridae, as well as the intralacustrine evolution

of a species flock, palaeolake Tiberino has been considered as a truly ancient lake, similar to the present-day lakes Ohrid, Biwa and Titicaca ("palaeo-ancient lake" [37–39]).

Together with the facies analysis, the oligotypic ostracod assemblages of the Cava Nuova section allow reconstruction of the physical and chemical palaeoenvironmental parameters.

The lowermost section (from 0 to 23 m, Interval 1), characterized by fine-grained sediments (clay and silt), indicates very low water energy and the prevalence of particles settling process [35]. The *Caspiocypris* group and *C. (N.) neglecta*, which prefer cool clayey floors, constitute the assemblages of this Interval 1. These taxa tolerate very low salinity conditions (0.5–6‰ [38]), whereas *C. (N.) neglecta* can also sustain low oxygen values, and prefer alkaline to weakly acid pH [40,41]. The ecology of ostracods indicates that the lake waters were probably moderately to well oxygenated, showing low salinity (between 0.5 and 6‰), neutral pH and low concentration of $CaCO_3$. As the grain size and fossil record are steady throughout the lower section, stable and long-lasting deep lacustrine conditions can be assumed. The inner part of the lake was characterized by cool and relatively deep waters (>40 m [40] or >50 m [42]) and by a silty to clayey floor.

Deposits in the intermediate section (from 23 to 38 m, Interval 2) show a coarsening-upward trend, accompanied by a sharp decrease in number of ostracod specimens. This is interpreted as a transition upwards to shallower depths, with higher water energy and a coarse-grained (sandy/silty) floor. These data are interpreted in terms of the shallowing trend and progradation of coastal systems, which presumably controlled the ecology of benthic forms [35].

Finally, in the uppermost section (from 38 to 44 m, Interval 3), the coarsening-upward trend proceeded, with deposits becoming prevailingly sandy and showing a high content of plant remains; all the data point to an increase in energy and reduction in depth. Deposits are associated with a wave-influenced, lacustrine coastal environment. The fossil content is here dominated by *C. torosa*, typically related to transitional waters, shallower depths and/or a sandy floor. As in the lower section, other chemical and physical parameters (water temperature, oxygen content, salinity, pH and concentration of $CaCO_3$) seemed to be steady throughout Intervals 2 and 3.

## 3. Materials and Methods

The freshwater ostracods considered herein come from 43 samples collected throughout the 45-m-thick CND section. The samples (each 500 g of sediment) were treated with hydrogen peroxide and water to remove the clay particles, and then filtered using a Satylon retina with a mesh of 63 μm. The washing residue was dried and observed under the stereomicroscope (Nissho Optical NSK). Valves with perforations were separated and metalized with Au-Pd for scanning electron microscopy (SEM, type Philips 515). The valve morphologies and the perforations were acquired, and for all the holes both maximum diameter (DM) and minimum diameter (Dm) were measured. Data on perforation sizes were compared with those reported in the literature [23,26,27,29].

## 4. Results

### 4.1. Total Abundance of Ostracods and Distribution of Microborings

The ostracod abundances (Figure 3, Table 1) refer to numbers of disarticulate valves, because entire carapaces were rarely found. The total number of valves recovered in all the samples is 2074, while the number of the valves in the 20 samples containing predated specimens is 1167 (Table 1). A total of 88 predated valves were collected, 5 of which were destroyed during the preparations for SEM analysis. From 0 to 24 m, the assemblages are composed 90% of *Caspiocypris* group specimens and only 10% of *C. (N.) neglecta*. Only rare *Caspiocypris* spp. specimens are found from 24 to 40 m, while from 40 to 43 m *C. torosa*, a species characteristic of a coastal lake environment dominates the assemblages.

**Table 1.** Ostracods of Cava Nuova, taxa recovered in assemblages and typology of predate valves. LV = left valve; RV = right valve; m = male; f = female; j = juvenile.

| | | Species | | | | | | | | | Total Per Sample | % Predated Per Sample |
|---|---|---|---|---|---|---|---|---|---|---|---|---|
| Sample | Number of Valves/Predated Valves | *Candona (N.) neglecta* | *Caspiocypris basilicii* | *Caspiocypris tiberina* | *Caspiocipris perusia* | *Caspiocypris tuderis* | *Caspiocypris posteroacuta* | *Cytherissa lacustris* | *Cyprideis torosa* | Juveniles Not Identifiable | | |
| 4 | valves | 5 | 33 | 10 | 9 | 0 | 0 | 0 | 0 | | 57 | 1.8% |
| | predated | | 1LVm | | | | | | | | 1 | |
| 5 | valves | 3 | 25 | 17 | 8 | 6 | 0 | 0 | 0 | | 59 | 5.1% |
| | predated | | 1LVm; 1RVm | 1LVm | | | | | | | 3 | |
| 6 | valves | 2 | 38 | 16 | 8 | 18 | 0 | 0 | 0 | | 82 | 4.9% |
| | predated | 2RVf | 1RVf | 1RVf | | | | | | | 4 | |
| 7 | valves | 2 | 35 | 14 | 9 | 14 | 0 | 0 | 0 | | 74 | 5.4% |
| | predated | 1RVf | 2RVf | | | 1LVm | | | | | 4 | |
| 8 | valves | 3 | 31 | 8 | 8 | 9 | 0 | 0 | 0 | | 59 | 3.4% |
| | predated | | 1RVf; 1LVm | | | | | | | | 2 | |
| 9 | valves | 2 | 27 | 12 | 12 | 10 | 0 | 0 | 0 | | 63 | 4.8% |
| | predated | 1RVf | 1RVf; 1LVf | | | | | | | | 3 | |
| 10 | valves | 2 | 26 | 6 | 6 | 4 | 0 | 0 | 0 | | 44 | 9.1% |
| | predated | 1LVm1LVf | 1RVf; 1LVm | | | | | | | | 4 | |
| 11 | valves | 3 | 19 | 11 | 6 | 5 | 0 | 0 | 0 | | 44 | 9.1% |
| | predated | | 1RVf | 1RVf | 1LVf | 1RVf | | | | | 4 | |
| 12 | valves | 2 | 21 | 5 | 3 | 7 | 0 | 0 | 0 | | 38 | 7.9% |
| | predated | | 2RVf; 1LVf | | | | | | | | 3 | |
| 13 | valves | 0 | 25 | 17 | 0 | 29 | 0 | 0 | 0 | 29 | 100 | 10.0% |
| | predated | | 1RVm; 1RVf 2LVf; 1LVj | 2RVf | | 1RVf; 2RVj | | | | | 10 | |
| 14 | valves | 0 | 28 | 30 | 3 | 20 | 0 | 0 | 0 | | 81 | 2.5% |
| | predated | | 1RVf; 1LVf | | | | | | | | 2 | |
| 15 | valves | 0 | 6 | 3 | 0 | 4 | 0 | 4 | 0 | | 17 | 17.6% |
| | predated | | 2RVf | | | | | 1LV | | | 3 | |
| 16 | valves | 1 | 12 | 1 | 2 | 18 | 0 | 0 | 0 | | 34 | 17.6% |
| | predated | 1RVf | 1RVf; 1LVf 1LVm | | | 2RVf | | | | | 6 | |
| 17 | valves | 1 | 24 | 15 | 1 | 6 | 0 | 0 | 0 | | 47 | 17.0% |
| | predated | 1RVf | 1LVf; 3RVf | 2LVm | | 1RVf | | | | | 8 | |
| 18 | valves | 0 | 23 | 14 | 5 | 8 | 0 | 0 | 0 | | 50 | 16.0% |
| | predated | | 3RVf; 2LVm | 2LVf | 1LVm | | | | | | 8 | |
| 19 | valves | 0 | 9 | 11 | 3 | 3 | 1 | 0 | 0 | | 27 | 7.4% |
| | predated | | 1LVf | 1LVf | | | | | | | 2 | |

**Table 1.** *Cont.*

| | | | | | | Species | | | | | | |
|---|---|---|---|---|---|---|---|---|---|---|---|---|
| Sample | Number of Valves/Predated Valves | Candona (N.) neglecta | Caspiocypris basilicii | Caspiocypris tiberina | Caspiocipris perusia | Caspiocypris tuderis | Caspiocypris posteroacuta | Cytherissa lacustris | Cyprideis torosa | Juveniles Not Identifiable | Total Per Sample | % Predated Per Sample |
| 20 | valves | 0 | 18 | 10 | 4 | 7 | 11 | 0 | 0 | | 50 | 20.0% |
| | predated | | 2LVm; 3LVf1RVf | | 1RVf | | 3LVm | | | | 10 | |
| 21 | valves | 0 | 10 | 14 | 3 | 3 | 4 | 0 | 0 | | 34 | 2.9% |
| | predated | | | 1LVf | | | | | | | 1 | |
| 22 | valves | 0 | 24 | 25 | 16 | 7 | 3 | 0 | 1 | | 76 | 7.9% |
| | predated | | 1RVf; 1LVm 3LVF | 1RVf | | | | | | | 6 | |
| 25 | valves | 0 | 53 | 15 | 25 | 4 | 28 | 0 | 6 | | 131 | 3.1% |
| | predated | | 1RVf | 1RVf; 2LVf | | | | | | | 4 | |
| | Total per species | 26 | 487 | 254 | 131 | 182 | 47 | 4 | 7 | 29 | 1167 | 7.6% |
| | | 8 | 50 | 15 | 3 | 8 | 3 | 1 | 0 | 0 | 88 | |
| | % predated per species | 31% | 10% | 6% | 2% | 4% | 6% | 25% | 0% | 0% | | |
| | Distribution of predated valves | 6RV,2LV 1m,7f | 27RV, 23LV 12m,37f,1j | 6RV, 9LV 3m,12f | 1RV, 2LV 1m,2f | 7RV, 1LV 1m,5f,2j | 3LV 3m | 1LV | | | 47RV,41LV 21m, 63f,3j | |

The abundance within samples is highly variable, from a minimum of 17 valves (sample CND 15) up to a maximum of 92 valves (sample CND 13). The abundances indicate an increasing trend from sample CND 0 to sample CND 6, and a progressive decrease until sample CND 12, followed by an increase up to sample CND 14. Sample CND 15, characterized by common iron oxides, contains only few specimens of *C. lacustris* registering an abrupt reduction of ostracod population. From sample CND 16 up to sample CND 25, the total abundances fluctuate, from a minimum of 27 specimens to a maximum of 131 specimens. From sample CND 26 to sample CND 38 until the associations become considerably impoverished, and before rising sharply in the sample CND 42 where they reach values of more than 300 specimens, mainly pertaining to *C. torosa*.

Regarding the percentages of the valves predated in relation to the total number of valves for each sample, from sample CND 4 to sample CND 9, where the number of valves is higher, the number of predated valves is very low (Figure 3, Table 1). From the sample CND 10 to sample CND 13, the number of predated specimens increases, while in sample CND 14, where the number of ostracods increases, the percentage of predated valves decreases. From sample CND 15 to sample CND 18, a progressive increase of ostracods appears in phase with the development of predation. In the sample CND 20, the highest percentage of predated valves is observed (20%), with a low total abundance (50 valves), which could be associated with a peak of abundance of predators.

*4.2. Shape and Morphology of Predation Holes*

The ostracod carapaces are built by the calcification of the outer of two layers of epidermal tissue, termed the outer lamella; the inner lamella is only partially calcified. The SEM analysis, performed on the best-preserved specimens with single and/or multiple perforations, allowed identification of different shapes of holes that were divided into six types on the basis of their measures, contour and shape (Figure 5).

- Type I (Figure 5a). Single hole with dimensions over 60 μm; truncated conical hole (symmetric paraboloid) with well outlined walls; evident traces of excavation, with special imprints (teeth or radula scratching) on the outer chitinous layer to reach and perforate the central calcitic layer. Referrable to ichnospecies *Oichnus paraboloides* Bromley, 1981.
- Type II (Figure 5b). Single perforation with circular to subcircular contour, dimension less than 50 μm; inner walls not well defined and hole eccentric with respect to the perforation of the outer chitinous layer. Referable to ichnospecies *Oichnus gradatus* Nielsen and Nielsen, 2001.
- Type III (Figure 5c). Perforation formed by two adjacent holes, comparable with ichnospecies *Dipatulichnus rotundus* Nielsen and Nielsen, 2001 (white arrow) and two irregular, partly overlapping holes (black arrow), which affected the outer chitinous layer and the central calcitic layer (not penetrative), referrable to a failed attack.
- Type IV (Figure 5d). Single large hole, over 80 μm, with vertical walls, associated with multiple and single traces of scratching. The external chitinous layer appears to be less thick, and traces of scratching seem to have affected only this part. Referrable to ichnospecies *Oichnus gradatus* Nielsen and Nielsen, 2001.
- Type V (Figure 5e). Large single hole, higher than 80 μm, with little marked vertical walls; the perforation crossed the chitinous outer layer and the central calcitic layer, while the chitinous inner layer was not perforated. Referrable to ichnospecies *Oichnus simplex* Bromley, 1981 (not penetrative).
- Type VI (Figure 5f, Figure 6m). Surface valve (left male of *C. tiberina*) covered by many sub-circular perforations and polyhedral scars. The subcircular type shows evidence of holes that reach through to the interior (referrable to *Oichnus simplex*), while the polyhedral etching scars do not seem to go over the central calcitic layer. Some of these scars could be attributable to fixichnia, i.e., to traces left by sessile organisms when they are anchored to a rigid substrate [18,43].

Types I–V appear to be the most common examples of perforations, many of which coexist on the same valve surface.

The best examples of microborings (Figure 6) evidence the presence of at least five ichnospecies: *Oichnus paraboloides* (Figure 6a,g), *Oichnus simplex* (Figure 6b,f) *Oichnus gradatus* (Figure 6c,e,i), *Oichnus* cf. *ovalis* (Figure 6l), and *Dipatulichnus rotundus* (Figure 6h).

It is important to highlight the presence of several traces of grasping, left by the erosion of the apparatus teeth (radula) (Figure 6d,e,l), and many examples of perforations with partially modified contours compared to the type shape of the ichnospecies (i.e., Figure 6l). This particularity could be due to the morphology of the valve, in fact in the case of a strongly arched shell, the regular action of the radula is hindered, which leads to an atypically formed hole [22]. The grasping traces and the single traces of scratching (Figure 5e) generally converge towards the boring, and are present in both adult and juvenile specimens. Figure 7 shows an example of scratching traces that converge in an incomplete predation, developed on a right female valve of *C. basilicii*.

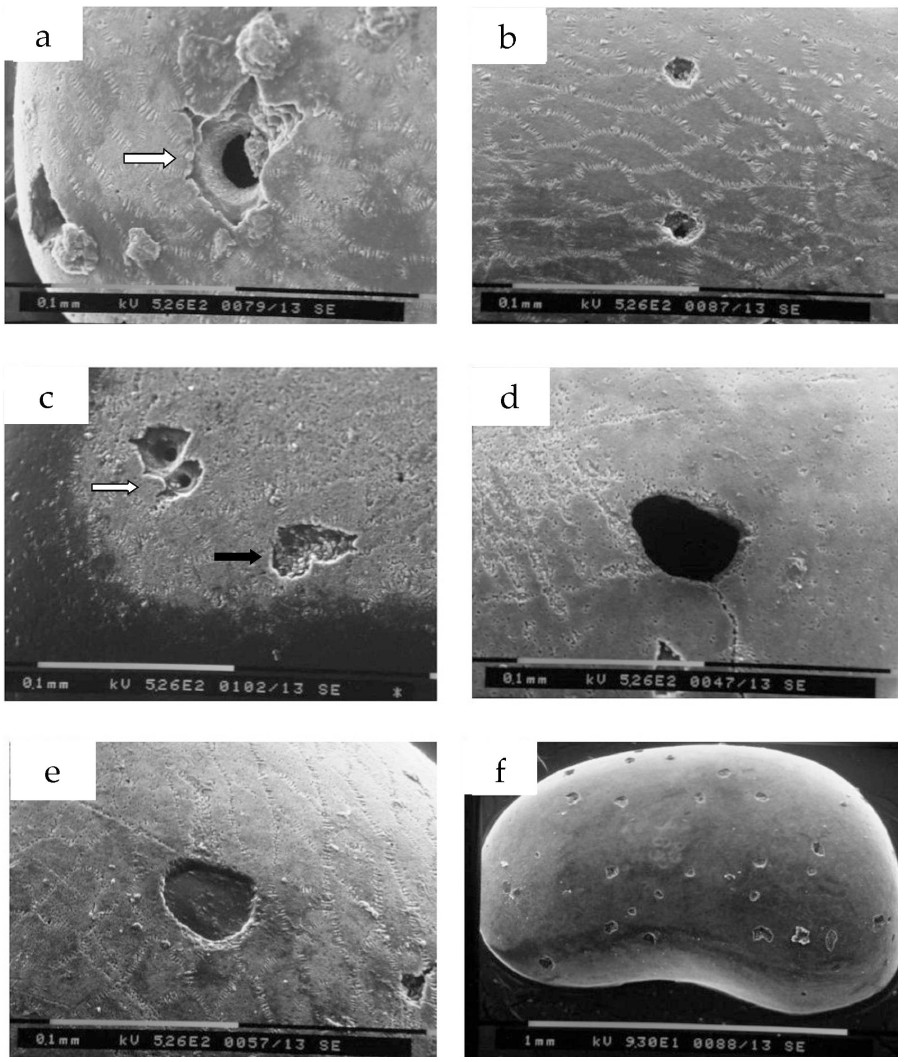

**Figure 5.** Examples of different type of perforations. (**a–e**) Scale bar = 100 μm. (**f**) Scale bar = 1 mm.

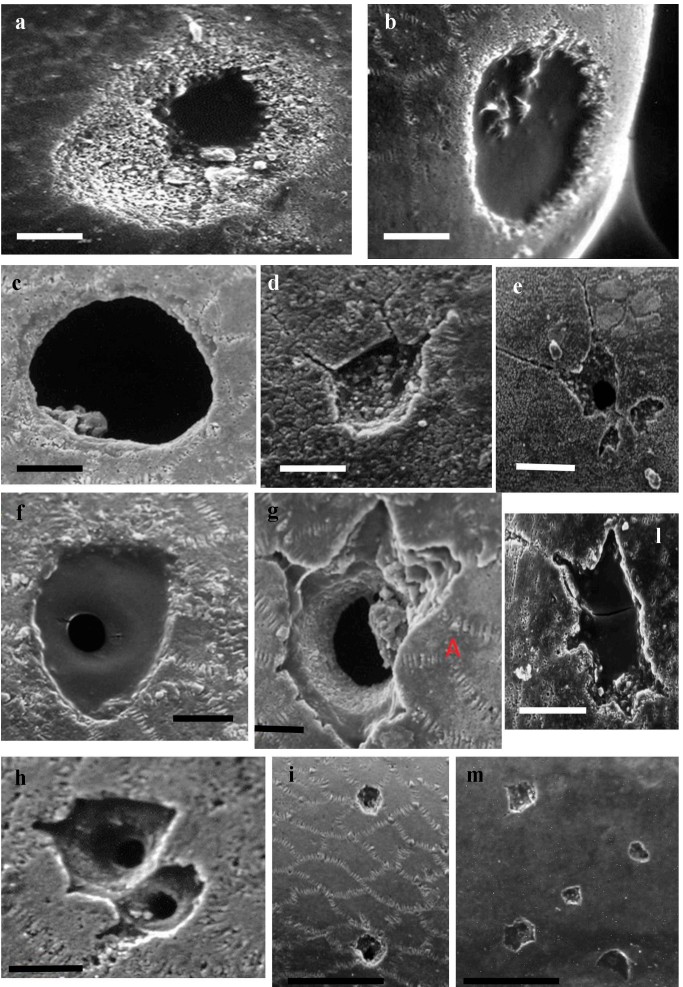

**Figure 6.** Examples of microborings. (**a**) *Oichnus paraboloides,* scale bar = 50 μm. (**b**) *Oichnus simplex* (not penetrative), scale bar = 50 μm. (**c**) *Oichnus gradatus*, scale bar = 50 μm. (**d**) Failed penetration with traces of rasping, scale bar = 50 μm. (**e**) *Oichnus gradatus* with traces of rasping and, in the lower right corner, of a special teeth of radular apparatus, scale bar = 50 μm. (**f**) *Oichnus simplex* (partly penetrative), scale bar = 50 μm. (**g**) *Oichnus paraboloides*, scale bar = 50 μm. (**h**) *Dipatulichnus rotundus,* scale bar = 50 μm. (**i**) Two holes of *Oichnus gradatus,* scale bar = 100 μm. (**l**) *Oichnus* cf. *ovalis* with traces of rasping around the scars, scale bar = 50 μm. (**m**) Scars with polygonal outlines (possible anchorage traces?), scale bar = 100 μm.

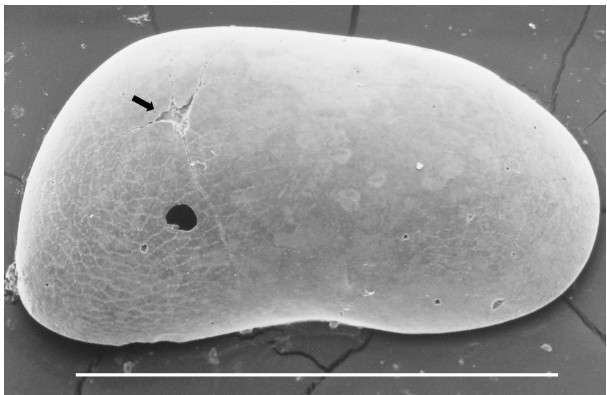

**Figure 7.** Examples of scratching traces (black arrow). Right female valve of *C. basilicii*, scale bar = 1 mm.

## 5. Discussion

The new data about valve bioerosion (mainly predation) have allowed broadening the knowledge of the microboring phenomenon in freshwater ostracods, and in particular to define:

(1) the frequency of predated valves and the palaeoenvironmental conditions under which predators developed, (2) the type of valves predated (right or left) and in some cases the sex of the specimens, (3) the types of predation (single or multiple), (4) attribution to ichnospecies, and finally, (5) the dimensional (size) difference with microboring ichnotaxa known from the literature. Unfortunately, no specific information about the candidate predators can be provided, and only limited hypotheses can be advanced.

### 5.1. Frequency and Type of the Predated Valves

The 20 samples containing predated valves were collected between 4 and 25 m (Figures 2 and 3, Table 1), where reddish, massive to slightly laminated clays with abundant iron oxides are the typical deposits. Specimens with evident marks of perforation are observed in sample CND 4, and the types of perforations are highly variable, both in shape and size. Most likely, some of these could be due to abiogenic phenomena of mechanical erosion, occurring post-mortem on the lake bottom, while others can be attributed to biogenic predation activity.

As predated valves only occur in Interval 1, the very low-energy deep-lake palaeoenvironment, and its clayey–silty floor, was the only one that was favorable to the presence and development of potential predators. According to age constraints (see Section 2.1), and to hypotheses on sedimentation rates [35], these environmental conditions were steady through a relatively short time interval (80–160 ka).

Likewise, this is the time interval to which all considerations of prey–predator relationships should be referred, including peaks in abundance of both ostracods and predators.

Perforations were identified on both right and left valves: 47 right valves and 41 left valves (Table 1). Among the 88 perforations (observed with SEM), 14 valves showed multiple traces of predation attacks of variable size and shape spread over the entire surface (Figure 8), and at least another 15 specimens had 3 or 4 perforations on the valves.

The female right valves (45) are numerically more common, while the male right valves (2) are subordinate (Table 1). Apparently, the data indicate that the right valves are the choice of predators, but this phenomenon is at present unclear. Among the predated adults, the female specimens are numerically higher. A possible explanation of this fact could be connected with the characteristic, typical of females, of carrying the eggs inside the carapace. Both the right and left female valves of *C. basilici* and *C. tiberina* appear heavily predated upon and therefore the hypothesis of attack against specimens capable of providing more nourishment, indicates that the choice by the predator is targeted. The research is still open, and many aspects will need to be further investigated.

The species *C. lacustris*, although very rare and present only in two samples (CND 15 and CND 16), provided one specimen with perforation.

An interesting fact emerges from the relationship between the number of potential prey and the frequency of predations. Analyzing the percentages of the valves predated in relation to the total number of valves for each sample, it can be seen that in the basal portion of the section (from sample 4 to sample 9), where the number of valves is higher, the number of predated valves is very low (Figure 3, Table 1).

On the other hand, when the number of valves decreases, the number of predated valves increases.

This trend leads one to suppose a negative relationship between the prey and the predators: low predation activity would seem to correspond to a maximum of available prey, and vice versa. A situation of this type could be indicative of a time-lag between the reproductive cycles of prey and predators [44].

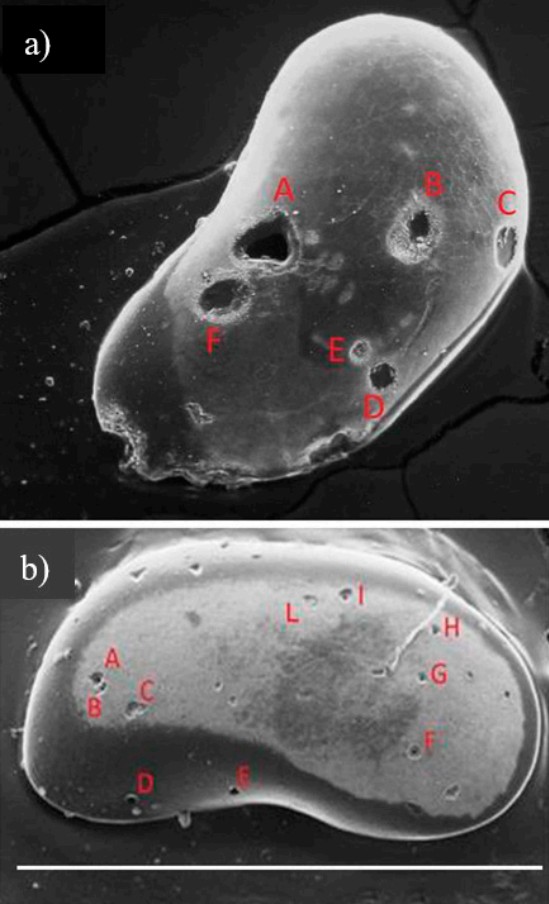

**Figure 8.** Examples of specimens with multiple perforations. (**a**) *Caspiocypris basilicii*, female right valve in external view, with evident abrasions along the anterior margin. Scale bar = 1 mm. (**b**) *Caspiocypris tiberina*, female right valve, external view. Scale bar = 1 mm. Red letters indicate the holes measured.

*5.2. Position of Microborings*

Most of the holes reported in the literature on the *C. torosa* valves [28] are positioned in the central area, near the inner muscle plaque. Nonetheless, in other cases, the position of the holes is highly variable, with a higher frequency in the anterior area of the valve [29], and the same microboring ichnogenus can be found in very different positions, apparently not related to the points of weakness of the valve.

In the studied specimens, the sites of predation are dispersed over the whole surface of the valve (Figure 9), and in one case they were also observed along the periphery of the valve (Figures 8a and 9e). No preferred perforation sites were recovered, but we noted that the largest ones were in the central-ventral area of valve, evidencing a favorite point of attack by predators. The only valve of *C. lacustris* affected by predation shows small holes and scratching dispersed along the periphery (Figure 4d, Figure 9c). The positions and the dimensions of microborings (Figure 9) clearly show two different types of predation, on the basis of size, and consequently at least two types of predators. The smallest ones, distributed over the whole surface, could have been produced by one kind of predator of relatively small size, while the large ones can be probably related to a larger predator.

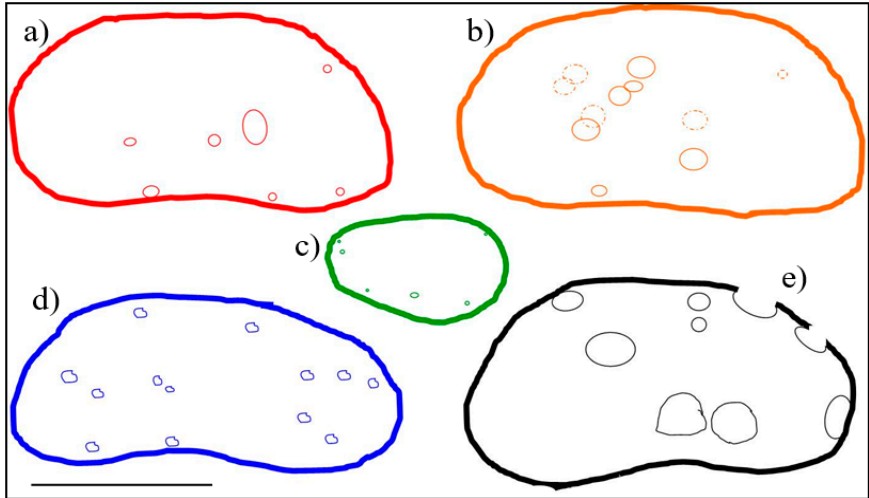

**Figure 9.** Profiles of the Cava Nuova valves and distribution of microborings. The dashed outline indicate the failed/or non penetrative predations. Scale bar = 500 µm. (**a**) *Caspiocypris basilicii*, female left valve. (**b**) *Caspiocypris tiberina*, female left valve. (**c**) *Cytherissa lacustris*, left valve. (**d**) *Caspiocypris tuderis*, female right valve. (**e**) *Caspiocypris basilicii*, female right valve.

### 5.3. Size Comparison of Ichnotaxa

The size of the ichnospecies (DM and Dm, external and internal diameters respectively, or maximum and minimum diameters) reported in [23,26–29] and the dimensions of our specimens were compared, with the aim of recognizing any similarities.

As reported for *O. paraboloides*, the diameters are variable from 50 to 500 µm [23,27]. Our measures of *O. paraboloides* indicate a maximum diameter (DM) variable from 18 to 143 µm, and the minimum diameter (Dm) variable from 4 to 95 µm. Our dimensions are included between the minimum and maximum values reported for the ichnospecies.

In the case of *O. simplex*, the holes reported [23,27] have a diameter variable between 25 and 125 µm. The diameters of the holes are 25 to 80 µm in euryhaline and freshwater ostracods, while on the valves of marine ostracods, they are between 100 and 150 µm [28]; in another case study, the dimensions were variable from 40 to 100 µm [29]. Our dimension are variable between 3 and 89 µm and are partially included in the range of variability reported for the ichnospecies, but we also observed some holes with opening values lower than 25 µm.

The ichnospecies *O. gradatus*, typically elliptical borings with a longest diameter of 90 µm and a shortest diameter of 75 µm, to a narrow internal width, c. 30 µm [26], in our case has a maximum diameter (DM) variable between 27 and 38 µm, and a minor diameter (Dm) variable between 9 and 16 µm. In the case of *O. gradatus*, the sizes are definitely smaller.

*O. ovalis* is a rhomboidal form with side to side widths averaging 120 µm, and borings do not appear to fully penetrate the substrate [26]. The sizes of the study specimens indicate the maximum diameter (DM) ranging from 98 to 134 µm and the minor diameter (Dm) ranging from 18 to 54 µm, and appear comparable with the data reported in the literature.

A particular hole with a heart-shaped opening, named *Oichnus* sp. [26] with a tip-to-tip length of 130 µm and a width at its broadest section of 120 µm, was also found in our specimens (Figure 8a, red letter A) with comparable dimensions. Remarkably, this particular morphology, previously reported from marine deposits [26], has now also been found in lake deposits.

The new data collected on deep lake ostracods contribute to enlarging the size range of the ichnotaxon *Oichnus* and also evidence the existence of a morphological convergence of hole shape, although the predators are probably different.

If the diameter of the boreholes is closely correlated with the size of the predators [22], then this information highlights the presence, inside this early Pleistocene deep lake environment, of small predators able to generate microborings of different sizes and shapes comparable to those produced by marine invertebrates.

The different sizes of drillholes could either be related to juvenile or adult stages of the same predator, or to distinct predators. At the present time of research, and lacking fossil remains of possible predators, it is impossible to discern between the two possibilities.

## 6. Conclusions

The ostracods affected by predation and intense microboring represent a very interesting population of confined taxa that carried out their whole life cycle on a relatively deep lake floor (more than 40–50 m) with low availability of $CaCO_3$ and low environmental energy. Furthermore, the size of drillholes, in some cases smaller than those known in the literature, indicate that the predators were of small dimensions and that they drew nourishment from the depauperate population of ostracods.

It has been noted [44] that the low correlation between density of individuals and intensity of predation may result from the time-lag between the population maxima of predators and those of their prey, and this situation was also detected in the studied section.

At present, we are not able to recognize what kind of predators attacked the ostracods, but the possibility that they were turbellarids [21] or other freshwater invertebrates is becoming a plausible hypothesis. The disappearance of predated valves, concomitant with the increased environmental energy (from 23 m, Figure 3), allows us to hypothesize that environmental conditions strongly influenced the development of predators, although the ostracofauna still lived on the lake floor. This hypothesis seems to be confirmed by the absence of predated valves in the samples at the top of the section, which is dominated by sandy sediments and very rich in ostracods (mainly *C. torosa*).

Finally, the discovery of predated ostracods in the deep lake environment of Cava Nuova, belonging to the western Tiberino basin, represents an extremely important datum because in the late early Pleistocene deposits of Arquata (belonging to the eastern Tiberino basin) referred to a relatively shallower lacustrine environment [45] the rich ostracod fauna did not show predated specimens.

We hope that the current research on lacustrine deposits, found at a drilling site in the sector belonging to the western Tiberino basin, provides further data on the phenomenon of predation on ostracods and increase this particular line of research.

**Author Contributions:** Conceptualization, A.B. and R.B.; samples preparation, F.P. and M.R.; investigation, A.B., R.B., F.P. and M.R.; data curation and analysis, A.B., R.B., F.P. and M.R.; writing—original draft preparation, A.B. and R.B. All authors have read and agreed to the published version of the manuscript.

**Funding:** This research received no external funding.

**Acknowledgments:** We acknowledge the technician Luca Bartolucci for assistance during the SEM analyses. We are grateful to the three anonymous reviewers for the punctual revision and for the stimulating discussions that improved the manuscript.

**Conflicts of Interest:** The authors declare no conflict of interest.

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
