# Peer review of "Evidence of Predation on Early Pleistocene Freshwater Ostracods (Umbria, Central Italy)"

_geosciences, doi:10.3390/geosciences10100416_

Round 1
Reviewer 1 Report
As the authors state, almost no work has been done on drillholes in ostracods, modern or ancient, despite ostracods' being significant members of biological communities and important in biostratigraphy. Evidently we are missing something. Similarly, very little work has been done on bioerosion of any kind in freshwater environments, so this piece of research is of interest in that area of research as well.
Just a few comments; the paper really needs few!
1. The significance of this research in freshwater bioerosion is understated by the authors. This should be stated outright in the abstract, and given attention in the Introduction with a few citations to works that summarize that literature. This will bring more attention to the current research.
2. The English needed work... I offer the attached annotated manuscript with corrections and suggestions for improvement.
Incidentally, I am not very happy with the phrased "predated valves", because "predate" is more naturally a verb with an entirely different meaning in English ("to date from an earlier time"), but this is hardly the first paper to use phrases like this, and the more natural "preyed on" does not lend itself to formal constructions. So let it stand.
3. In lines 56-57, several ichnogenera are referred to without date. These should be cited, as this is a taxonomic paper.
4. In line 410, does "side width" mean "valve thickness"? That would probably be a better phrase (not "shell thickness", which I wrote in the annotated ms).

Author Response
Dear Reviewer
thank you so much for the careful review of the manuscript and for your valuable advice and comments.
We accepted almost all the modifications you proposed, particularly the ones regarding English. All the changes made to the text are highlighted in the uploaded version, and we also provide a point-by-point reply in the cover letter. You can find there all the details.
Here, some major reply to your comments:
- Abstract has been modified, by adding the sentences: “Although drillholes in modern or ancient ostracods are known, the record is relatively scarce if compared to other taxa, and mainly referred to marine environment. Moreover, less is known about perforated ostracods, and more generally, about bioerosion in freshwater environments” (lines 14-17), and “also suggesting prey-predator relations through a relative short time interval (80-150 ka)” (lines 32-33). We hope these changes can help to enhance our research.
- The text have been modified mainly following the comments and corrections you provided in the annotated manuscript. Major changes are reported in the point-by-point reply, while minor changes (use of articles, spelling, capital letters, etc.) are directly highlighted in the revised version. thank you so much for your patience in revising our English.
- We agree, several ichnogenera are referred to without date. Lacking dates have been added as required
- "side width" was introduced by authors [26] to describe the average dimensions of the rhomboidal form in O. ovalis.
Best Regards
Angela Baldanza and co-authors
Reviewer 2 Report
this is a well performed study without any major flaws
images are prepared at a highl professional level including the SEM microphotographs
this references list appears to be complete and exhaustive
both conclusions and discussions are well supported and justified
the only one and only very minor issue is on the line 479: the crossed out number "11"
otherwise i havn´t found anything else and would like to wish authors good luck with looking for the culprits of these peculiar freshwater traces
Author Response
Dear Reviewer
thank you so much for the careful review of the manuscript and for having detected the typing error in the bibliography. We have taken steps to correct it.
Best regards
Angela Baldanza and co-authors
Reviewer 3 Report
Dear editors, dear colleagues,
You requested my expertise to review the article entitled “Evidence of predation on Early Pleistocene freshwater 2 ostracods (Umbria, central Italy)” by Baldanza and colleagues for publication in Geosciences.
This contribution provides an analysis of predation marks seen on valves of ostracods from freshwater deposits, corresponding to deep lake, of Early Pleistocene age outcropping in central Italy. Predation on ostracods is generally still poorly understood, especially in freshwater environments and even more in the fossil record. In that sense, this work is an important contribution. Modifications, remarks and questions are mentioned within the text, I did not check the references list. The text should be carefully revised to adjust grammar and spelling issues, for instance plural versus singular, predate/predated… An important work is however needed to re-organize the manuscript that mixes results and discussion in its current version. Several points need to be added both in the results and the discussion, see below, to have a complete view of the material and observed trends. Figures also need to be modified to provide appropriate information for the reader to follow. For these reasons, I suggest that it can be accepted for publication after major revisions.
Below are more detailed questions and remarks.
Introduction:
- The first part of paragraph ranging from line 54 to line 67 (highlighted within the manuscript file), dealing with the taxonomy of ichnogenera and ichnospecies, is of interest in describing the history of the classification of these genera as well as the current and adopted scheme. However, this part lacks of clarity with long sentences that should be divided, with some additional details. The final sentence of this paragraph (lines 65-67) as also dense in important information that are not clear. Clarifying these elements, without falling into too many details, would greatly help the reader.
- It is not clear whether the paragraph ranging from line 71 to line 76 corresponds to general observations issued from a body of literature or are results of the present investigation. It should be slightly rephrased to clarify this point.
- Lines 83 to 110: here three examples of predation on ostracods are reported, but the literature on this topic is much more abundant than that. Why are these three examples chosen? These examples correspond to marine conditions, it be much more significant for the present analysis and for the reader to briefly present the available information on predation in freshwater?
- As a whole, the content of the introduction is adequate to the topic discussed but it should be slightly re-organized and clarified to gather all elements on the classification of ichnotaxa in one paragraph, all elements of the current knowledge of predation on ostracods in another one, and so on. Owing that analyses on predation on ostracods in freshwater areas are few, it would be much more significant to described what is known for this type of environment rather than in marine areas, to highlight the many questions still pending and what the present analysis brings to the discussion.
Geological setting:
- In lines 169 and 170, is it a new stratigraphic constraint you propose? If yes, it would be interesting to get more details on the proxies used for calibrating the sections with which you are comparing the one you studied. If no, just a brief information on the proxies used would be important.
- In figure 3, it would be good for the reader to increase the size of the elements corresponding to gastropods, bivalves and leaves. I did not manage to find where plan remains occur in the lithological column: either the indication should be added or the legend should be deleted. Generally, all line drawing are very tiny in this figure: could at least the ostracod horizons be highlighted, as it seems that the one occurring around 20m in interval 1 corresponds to the peak of perforated valves? In this figure, it would be interesting to add a column with the distribution of all species listed in the text and a reference to this figure may be added in lines 186, 187. This would make it more easy to follow for the reader, including the part describing the successive paleoenvironments in lines 209 to 230.
- Figure 4: please modify the orientation of specimens shown in d, f, h to illustrate the ventral margin in the lower part. Specimen shown in d is not a right valve. Specimen shown in f is a right valve, please add this.
Results:
- Lines 248 to 250: it would help the reader if the samples that yielded predated valves were highlighted in Figure 3. Lines 248 to 250, it is not clear whether the information correspond to the total abundance of valves or only to the abundance of predated valves per sample. Similarly, an analysis of the pattern of predation in relation to the ostracod species involved is lacking, except if the sentence I commented in in lines 247, 248 is actually dealing with this question. It should also be important to precise if all samples record the same pattern of predation regarding the ostracod species. A last question: most of the specimens recovered are isolated valves, is there a pattern regarding which valves are rather attacked, right or left? or is the predation rather homogenous?
- Paragraph 4.1: according to the title of this paragraph, both total abundance and distribution of microborings should be discussed. However, it seems that only total abundance is discussed, in a unclear manner: as stated above, modifications of Figure 3 to add the distribution of species and to highlight the samples that produce ostracods and microbored valves would be very helpful to the reader. This paragraph needs to be reworked to integrate microboring distribution and to clarify the patterns.
- Paragraph 4.2: can the scratches observed on holes of type IV (line 180) rather be related to preservation and/or action of micro-encrusters following the death and deposition of the valves?
- Paragraph 4.2 again, lines 180, 181: is this boring type restricted to a certain taxa with thin valves?
- Paragraph 4.2, Type VI: several morphologies of holes are mentioned but not clearly visible on the photo in Figure 5, they should be marked to help the reader and this photo should maybe be enlarged.
- Line 296: Oichnus ovalis (please change cfr. into cf.) is mentioned as occurring on the studied specimens but it has not been listed, discussed and pictured in the description of the diverse types of holes (only pictured in Figure 6). This should be clarified.
Discussion:
- Table 1 (p. 11) contains purely descriptive elements that should be moved to the Results part. These elements were necessary when reading the part of the results, as shown by some of my previous comments. The results need to be reworked to integrate this table as well as the description of the information it provides. Similarly, the elements in lines 331-333 are purely descriptive and should be moved to the results part.
- Lines 333 to 335: the way this sentence is written is not clear, I assume that it is a deduction owing to the fact that this interval of the section is the only that provided predated valves? It should be clearly stated.
- Lines 336-339: are there differences in the predation on RV or LV among samples?
- Lines 346, 347: numbers should be given to illustrate this observation.
- Lines 356 to 361: this part is also purely descriptive and should be moved to the results part.
- Figure 8: more information are needed in the caption: species? labelling the specimens would allow to just refer to Fig.8E instead of “Figure 8 black profile” (line 382) for instance.
- Lines 380-389: is there a link between the position of the microborings on the predated valves and the ichnospecies?
- Lines 421-424: the different size may correspond to distinct and/or juvenile and adult predators ? Maybe a word can be added on that aspect.
Conclusion:
- Lines 438, 439: this information on sandy sediments and associated structures do not belong here but in the geological setting.
Author Response
Dear Reviewer
thank you for the careful review of the manuscript and for your valuable advice and comments. We agree the work needed to be partly reorganized, according with your suggestions and the comments of the other reviewer. We added the lacking dates in taxonomic references. As suggested, we check and revise the English.
All the changes made to the text are highlighted in the uploaded version, and we also provide a point-by-point reply in the cover letter. You can find there all the details.
Here, some major reply to your comments:
Introduction:
- Introduction has been divided in two parts, according with comments. The first paragraph deals to classification of ichnotaxa, the second (with its own headline) is focused on predation on ostracods (see uploaded version)
- Lines 72-93 (original line 54 to line 67, final sentence original lines 65-67) this part has been reorganized and rewritten (see uploaded version)
- Lines 93-99 (original from line 71 to line 76): this sentence is referred to the cited literature [23]. Anyway, this phrase has been rewritten (see uploaded version)
- Lines 223-246 (original lines 83 to 110): Unfortunately we have not found other literature examples of microboring on freshwater ostracods in a deep lake environment. There are data in the literature regarding predations on current ostracods living in temporary water pools, but in that case a predation that completely destroys the valves has been documented, and the predators are generally juvenile stages of aquatic insects. We selected the works that allowed us to compare them with brackish environments in which specimens of Ciprideis torosa were found, a species of fresh water that tolerates high salinities, which we also found in our section of Cava Nuova (obviously not predated!). We would be grateful if you have any specific papers that we can enter to implement this part. The work [27] was chosen, despite referring to a marine context, for the good correspondence between the forms described and those documented in our case study.
Geological setting:
- lines 418-429 (original lines 169 and 170: this age constraint has been proposed in [35], and derives from the proposal of [37,42] to use endemic ostracods of FBU as stratigraphic proxies. To highlight it, this part has been modified (see the uploaded version)
- Figure 3 has been modified according to comments.
- Figure 4: We choose this upside-down orientation for specimens in d, f, h to make images readable. When rotated, the position of the light degrades image quality. We agree specimen in d is a left valve (modified in caption), and the one in f is a right valve (information added in caption)
Results:
- Lines 613-614 (original lines 248 to 250): samples with predation have been highlighted in figure 3. The other details are now mainly reported in Table 1.
- Paragraph 4.1: the paragraph has been reorganized, now putting more attention on the samples with predations; data have been also synthesised in Table 1.
- Paragraph 4.2: The photos and drawings we consulted are those reported by Danielopol et al, 1986 (On the preservation of carapaces of some limnic ostracods: An exercise in actuopalaeontology. Hydrobiology 143: 143-157), concerning the traces of corrosion in the carapaces/valves produced first by microorganisms (bacteria/fungi) and then by chemical attacks. The drawings show traces of corrosion organized in a radial pattern, very different from those we have found.
- Paragraph 4.2 again: The valves of Caspiocypris spp. and of Candona neglecta are very thin, so much so that the impressions of the adductor muscles of the valves are clearly visible even in external lateral view. This fact is related to low value on calcium carbonate dispersed in lake waters.
- Paragraph 4.2, Type VI: Figure 5 has been enlarged
- Oichnus ovalis (“cfr.” modified in “cf.”) is pictured in Figure 6l. It has been mentioned as occurring on the studied specimens, and it has not been listed and/or discussed because it occurs only in one predated valve.
Discussion:
- Both Table 1 and original lines 331-333 have been moved in the Results (lines 570-571). Table 1 has been modified, to integrate the information required about distribution of predation among species, gender and right/left valves.
- Lines 673-675 (original lines 333 to 335): yes, it is a deduction owing to the fact that this interval of the section is the only that provided predated valves. To clarify, the sentence has been modified in: “As predated valves only occur in Interval 1, the very low-energy deep-lake palaeoenvironment, and its clayey-silty floor, was the only one that was favorable to the presence and development of potential predators”
- Lines 680-683: Unfortunately, at the moment this datum is missing, but it would be possible to extract it by checking the predated species in greater detail. Many observations have been made but unfortunately we are aware that others should have been made. The idea of reporting predations on freshwater ostracods has been underestimated by us for its real interest, but we will certainly try to increase the data and publish it soon.
- Lines 784-791: modified as follows: The female right valves (45) are numerically more common, while the male right valves (2) are subordinate (Table 1). Apparently, the data indicate that the right valves are the choice of predators, but this phenomenon is at present unclear. Among the predated adults, the female specimens are numerically higher. A possible explanation of this fact could be connected with the characteristic, typical of females, of carrying the eggs inside the carapace. Both the right and left female valves of C. basilici and C. tiberina appear heavily predated upon and therefore the hypothesis of attack against specimens capable of providing more nourishment, indicates that the choice by the predator is targeted. Research is still open and many aspects will need to be further investigated.
- Figure 8 (now Figure 9) and its captions have been modified (profiles labelled with letters and species reported). Citations in the text have been modified accordingly.
- Lines 843-852: Probably yes, but at moment we not are sure. Others observations on predate valves will be necessary and we hope to increase data with supplementary analyses on new predate valves that will be extracted from the new laboratory treatment of CND samples.
- Lines 900-903: The different size could either be related to juvenile or adult stages of the same predator, or to different predators. At the present time of research and lacking fossil remains of possible predators it seems a forcing. But we added a sentence.
Conclusion:
- Line 847 (original lines 438, 439): this part has been moved in the geological setting (lines 395-397)
Best Regards
Angela Baldanza and co-authors